# Yield Advantage and Economic Performance of Rice–Maize, Rice–Soybean, and Maize–Soybean Intercropping in Rainfed Areas of Western Indonesia with a Wet Climate

Erythrina Erythrina [1,*], Susilawati Susilawati [1], Slameto Slameto [1], Ni Made Delly Resiani [1], Forita Dyah Arianti [1], Jumakir Jumakir [1], Anis Fahri [1], Andy Bhermana [1], Asmanur Jannah [2] and Hasil Sembiring [1,3]

1  National Research and Innovation Agency, Jl. M.H. Thamrin No. 8, Central Jakarta 10340, Indonesia
2  Faculty of Agriculture, Nusa Bangsa University, Jl. KH Sholeh Iskandar KM.4, Bogor 16166, Indonesia
3  International Rice Research Institute—Indonesia Office, Jl. Merdeka No. 147, Bogor 16111, Indonesia
*  Correspondence: erythrina_58@yahoo.co.id; Tel.: +62-813-8178-0034

**Abstract:** Cereals–soybean intercropping is not well studied, despite the importance of these crops in increasing agricultural profitability and ensuring nutritional and food security in Southeast Asia. We compared different intercropping practices (IPs) with monocropping practices (MPs) for their yield and economic performance by small-scale farms without mechanization. The treatments were IPs of rice–maize, rice–soybean, or maize–soybean compared with MPs of rice, maize, or soybean as sole crops, across three provinces in the rainfed areas of western Indonesia with a wet climate. Our results show that the yield advantages using the land equivalent ratio of the IPs were 44% for rice–maize, 54% for rice–soybean, and 63% for maize–soybean compared to MPs. Rice equivalent yield, maize equivalent yield, and the gross margin under IPs were significantly higher per cycle than under MPs; IPs provided a substantially lower cost of production and of paid workers. Compared to just rice, there were additional net return gains of USD 160 and USD 203 ha$^{-1}$ per cycle under rice–maize and rice–soybean intercropping. Maize–soybean intercropping resulted in an additional net return gain of USD 153 ha$^{-1}$ compared to just maize. These results suggest there is considerable potential for small farmers to increase their yields and profits by intercropping in rainfed areas with a wet climate.

**Keywords:** rainfed; cropping systems; productivity; net income; food security

## 1. Introduction

Many diverse cropping systems have increased food production and farmers' incomes in Africa, India, and China [1–3]. One of these systems is intercropping, i.e., growing two or more crop species simultaneously in the same field during a growing season [4]. Efficient utilization of land resources, where scarcity of land compels farmers to grow many crops on a small piece of land, is one of the rationales for intercropping in traditional farming systems [5]. Recent research [6–8] has raised several concerns about the future sustainability of rice–maize cropping systems. Yield growth rates have slowed and reached a plateau in some significant rice-producing regions including Indonesia, representing a potential issue in meeting future rice demand, which is expected to increase. Due to its limited land for food production and high population, Indonesia is testing various methods in an attempt to increase the production of food crops, especially of rice, maize, and soybean [9–12]. Maize and soybean are the second and third most important strategic commodities after rice. Rice–maize intercropping systems are crucial in ensuring food security. Rice–soybean or maize–soybean combinations provide food and nutritional security to smallholders of rainfed lands. Thus, they may be considered suitable options for small farmers' food and livelihood security. Out of around 20.3 million food crop farmer households in 2018 [13], intercropping systems are applied in approximately 12.5% of upland rice households, 15.8% of maize households, and 23.2% of soybean households [14]. Intercropping has some

advantages for small farmers whose farm operations are labor-intensive and who use simple tools for cultivation on smallholdings [15].

The potential area of rainfed with a wet climate available for food crop development in Indonesia is 2.7 million ha, and such areas are primarily found in Sumatra, Java, Kalimantan, and Papua [16]. Climatologically, they are characterized by having >1500 mm year$^{-1}$ rainfall and several wet months, i.e., >7 months with rainfall of >100 mm month$^{-1}$. Due to a longer growing season, where multiple cropping is feasible, the capture of resources and yield are often improved through facilitation and niche differentiation in time and space [17]. Due to a longer growing season, the potential advantage of combining cereal–legume intercropping provides greater scope for minimizing the adverse impact of moisture and nutrient stress and improving system productivity. In terms of the time component crops take in the intercrop, cereal–legume intercropping is shorter when compared to the growing season.

Except for the soils in West Nusa Tenggara, East Nusa Tenggara, and East Timor (Figure 1), where the climate is relatively dry, the lands of the other areas have mainly developed under humid tropical conditions from acid sedimentary rocks. In general, Ultisols and Oxisols are dominant [18]. Oxisols and Ultisols are highly weathered, low-activity, freely draining soils with low content of weatherable minerals, low nutrient retention, and high leaching pressure. Therefore, upland soils are acidic with high aluminum (Al) saturation, low phosphorus, and low base saturation. Acidity and Al toxicity are the most critical agronomic problems [19,20].

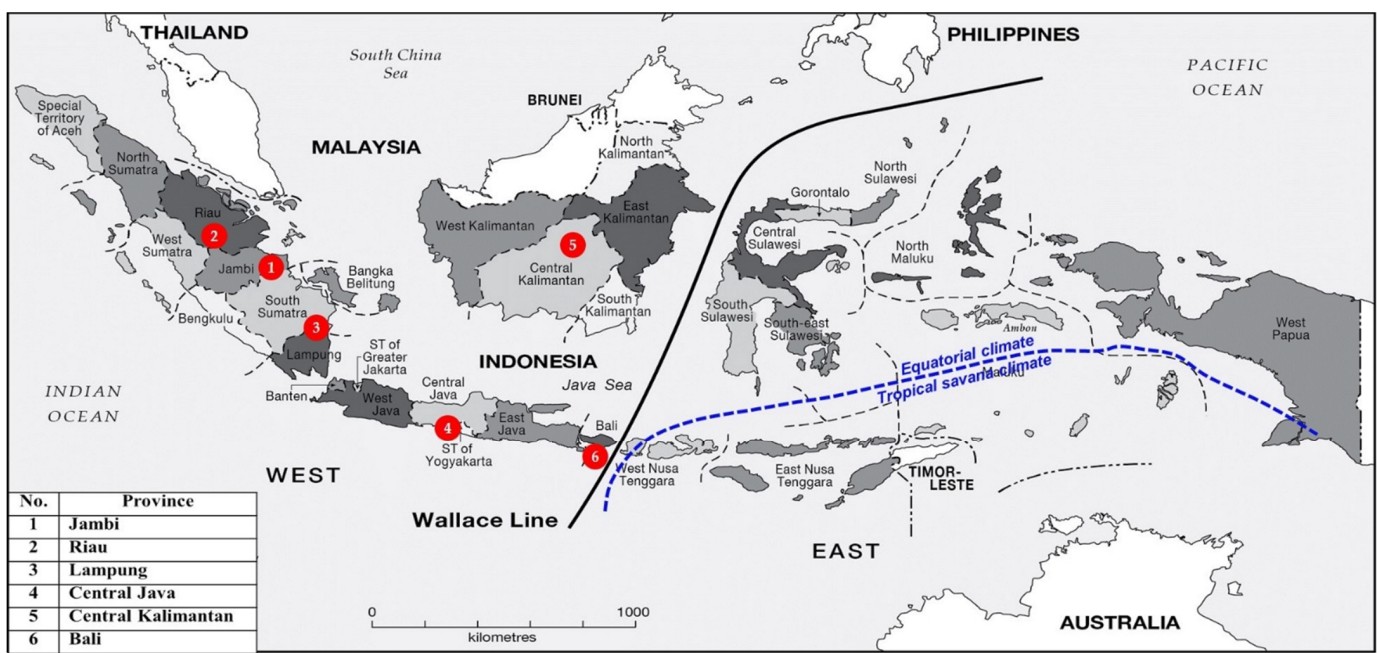

**Figure 1.** Map illustrating the six wet-climate provinces of western Indonesia designated as study areas.

Small family farms characterize farming in China and Southeast Asia; they are usually less than 1 ha in size, and farm income is an essential element of livelihoods in these regions [21,22]. To accelerate the adoption and dissemination of intercropping patterns, farmers must consider several factors, such as the planting density of the two mixed crops, the adaptation of high-yielding cultivars, increasing production costs, and yields and profits [23,24]. Few studies on rice–maize, rice–soybean, or maize–soybean intercropping have evaluated the increase in farmer land productivity and profitability, especially in rainfed areas of Southeast Asia [25,26]. It is increasingly rare to find intercropping studies in which farmers are directly involved, such as participatory demonstration plots related to labor productivity and cost productivity in actual conditions.

Different indices are used to assess crop yields, the competition intensity of the component crops, and the economic efficiency of intercropping practices compared with monocropping practices of sole crops [27–30]. In this research study, we used the gross margin and profit analysis under farmer participatory demonstration plots to assess the intercropping system's financial viability in addition to those indices. Therefore, we hypothesize that to increase the profit margin, intercropping practices may reduce the cost per unit output and increase labor productivity compared to monoculture practices. Here, we focus on rice–maize, rice–soybean, and maize–soybean intercropping systems as a case study. We aimed to assess different intercropping practices of rice–maize, rice–soybean, or maize–soybean compared to monocropping practices of rice, maize, or soybean as sole crops based on yield advantage and economic performance in the rainfed areas of western Indonesia with a wet climate.

## 2. Materials and Methods

### 2.1. Site Description

The study covered selected wet-climate provinces in western Indonesian engaged in rice–maize, rice–soybean, and maize–soybean intercropping in rainfed areas in 2019. Figure 1 and Table 1 show a map and details, respectively, of the farmer participatory demonstration sites in the six provinces designated as study areas.

**Table 1.** Location and farmer participatory demonstration sites (FPDSs) of rice–maize, rice–soybean, and maize–soybean intercropping systems in rainfed areas of western Indonesia with a wet climate, 2019.

| Province | District | Subdistrict | Village | FPDS | Coordinates |
|---|---|---|---|---|---|
| | | | Rice–maize intercropping | | |
| Jambi | Merangin | Pamenang Selatan | Tambang Emas | Pasundan | −2°12′16″; 102°22′11″ |
| Lampung | Tanggamus | Pugung | Banjar Agung | Karya Makmur | −5°20′46″; 104°48′39″ |
| Central Kalimantan | Kotawaringin Timur | Mentaya Hulu | Santilik | Santilik Bersinar | −1°57′5″, 112°37′52″ |
| | | | Rice–soybean intercropping | | |
| Riau | Kampar | Perhentian Raja | Hang Tuah | Melati Indah | 0°18′50″; 101°23′45″ |
| Lampung | Tanggamus | Bulok | Banjar Masin | Umbul Solo | −5°26′53″; 104°54′35″ |
| Central Kalimantan | Kotawaringin Timur | Mentaya Hulu | Santilik | Mantep Tani | −1°57′22″; 112°37′53″ |
| | | | Maize–soybean intercropping | | |
| Lampung | Central Lampung | Pubian | Payung Rejo | Sri Rejeki II | −5°5′52″; 104°53′5″ |
| Central Java | Pemalang | Ampel Gading | Tegalsari Barat | Rawa Bingung | −6°96′95″; 109°30′31″ |
| Bali | Tabanan | Selemadeg Timur | Tanguntiti | Subak Aseman IV | −8°31′25″; 115°2′44″ |

Data regarding the average monthly rainfall, and maximum and minimum temperature for the last ten years (2010–2019) were collected from climate stations in the study sites (Table 1). In Sumatra, Java, and Kalimantan, October/November coincides with the onset of the wet season for most areas, and the wet season continues until the end of June [31]. Figure 2 shows that for all study sites, the highest cumulative precipitation during the growing season was noted in Jambi at 1375 mm and the lowest in Bali at 1149 mm. The maximum temperature ranged from 25.5 to 35.4 °C, and the minimum temperature varied from 20.3 to 24.1 °C.

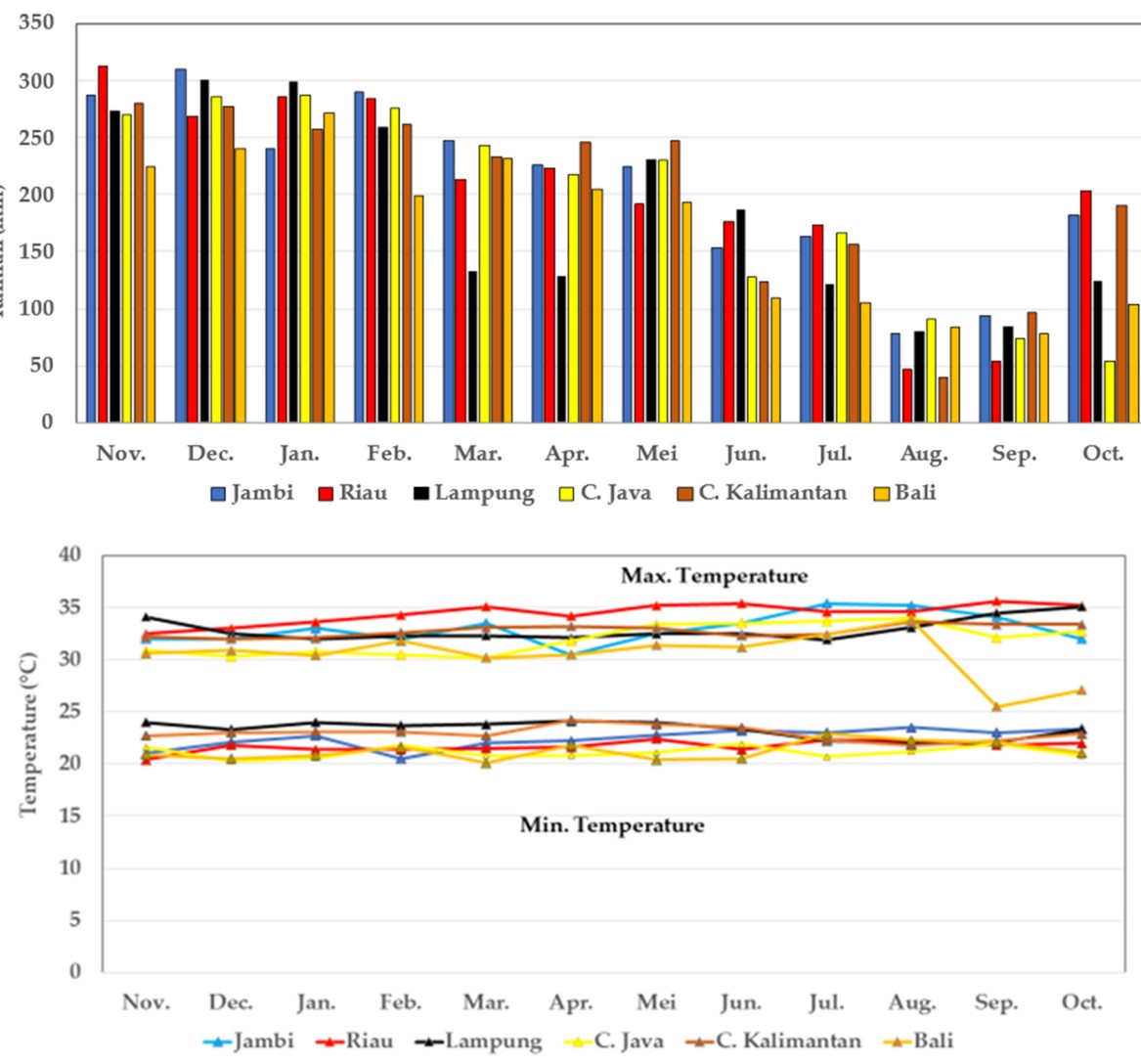

**Figure 2.** Average monthly rainfall, maximum and minimum temperatures for the period 2010–2019 in the study sites.

*2.2. Intercropping Practices (IPs) and Monocropping Practices (MPs)*

The selection of districts in the province is based on the broadest cropping pattern (rice–maize, rice–soybean, or maize–soybean) at the district level [13]. Furthermore, one sub-district was selected from the district, which had the same widest cropping pattern. The sub-district determined one village as the Farmer Participatory Demonstration Plot (FPDP) location. The treatments were intercropping practices (IPs) of rice–maize, rice–soybean, and maize–soybean compared with monocropping practices (MPs) of rice, maize, or soybean as sole crops. Each intercropping system of rice–maize, rice–soybean, and maize–soybean had been used by five cooperating farmers (blocks) randomly selected from the pool of farmer groups in each village and was designated as a replication in each intercropping. The farmer participatory demonstration sites (FPDSs) as on-farm experiments were arranged in a randomized complete block design using the one-farmer–one-block approach. The same intercropping pattern was used across three provinces (Table 1); thus, we had 15 cooperating farmers for each intercropping system or 45 cooperating farmers for rice–maize, rice–soybean, and maize–soybean intercropping. The field plot area for each intercropping system was 100 m$^2$, and each sole crop area was 100 m$^2$ for comparison. The same farmers grew the intercrops and sole crops at the same management level. Each cooperative farmer was given the recommended site-specific variety of rice, maize, and soybean seeds and

fertilizers for free. Newly released high-yielding varieties (HYVs) underwent performance testing to combine high yield potential, tolerance to acidic soils, and disease resistance [32]. The newly released varieties were Inpago-12 for rice, hybrid variety Nasa-29 for maize, and Biosoy-2 for soybean. The costs of different operations, including land preparation, planting, fertilizing, weeding, pesticide spraying, harvesting, and processing were recorded by field technicians with the assistance of the researchers. The mean data on the grain yield of each FPDS between IPs and MPs across the three provinces were replicated in this study.

*2.3. Cultural Practices*

As the main crop under rice–maize intercropping, rice is planted two weeks before maize planting. Rice is planted with three to four seeds hill$^{-1}$ and maize with one seed hill$^{-1}$. Soybean is grown about two weeks after rice under upland rice–soybean intercropping. Upland rice is planted as a main crop with three to four seeds hill$^{-1}$ and soybean is planted with two seeds hill$^{-1}$. As the main crop in the maize–soybean intercropping, maize planting is carried out two weeks earlier than soybean planting. The farmers sowed one seed for maize and two seeds for soybean. The number of rows, inter-row distance, intra-row distance, and land proportion under different IPs and MPs for all crops are shown in Table 2 and Figure 3. For each IP, the sole upland rice at 160,000 plants ha$^{-1}$, maize at 55,556 plants ha$^{-1}$, and soybean at 133.333 plants ha$^{-1}$ were maintained as MP controls. The fertilizer package comprised inorganic fertilizers of urea (45% N) and NPK (15:15:15). The fertilizer rates were 45 N, 22.5 $P_2O_5$, and 22.5 kg $K_2O$ ha$^{-1}$ for sole upland rice, 135 N, 45 $P_2O_5$, and 45 kg $K_2O$ ha$^{-1}$ for sole maize, and 37.5 N, 15 $P_2O_5$, and 15 kg $K_2O$ ha$^{-1}$ for sole soybean. In the intercropping, fertilizers were applied proportionately based on the mixed ratio to the sole plant population for main crops and intercrops separately. The rates of fertilizer application and plant spacing were deliberately chosen, as they represent what the local farmers can usually afford on average.

**Table 2.** The number of rows, distance between rows, distance in rows, mix ratio, and expected plant density under different IPs and MPs in drylands of western Indonesia with a wet climate.

| Crops | IPs | | | | | | MPs | |
|---|---|---|---|---|---|---|---|---|
| | Number of Rows | Distance between Rows | Distance in Rows | Mix Ratio | Plant Density | Average Growing Period | Planting Distance | Plant Density |
| | | (cm) | (cm) | (%) | ha$^{-1}$ | (days) | (cm) | ha$^{-1}$ |
| | | | | Rice–maize intercropping | | | | |
| Rice | 9 | 20 | 15 | 60 | 199,800 | 113 | 25 × 20 | 200,000 |
| Maize | 2 | 60 | 25 | 40 | 26,667 | 104 | 70 × 25 | 57,143 |
| | | | | Rice–soybean intercropping | | | | |
| Rice | 9 | 20 | 15 | 55 | 183,333 | 114 | 25 × 20 | 200,000 |
| Soybean | 5 | 30 | 15 | 45 | 100,010 | 84 | 30 × 25 | 133,333 |
| | | | | Maize–soybean intercropping | | | | |
| Maize | 2 | 60 | 25 | 36 | 24,000 | 105 | 70 × 25 | 57,143 |
| Soybean | 7 | 30 | 15 | 64 | 142,222 | 86 | 30 × 25 | 133,333 |

*2.4. Parameters Evaluated*

Different indices were used to assess the crop yields, the competition intensity of the component crops, and the economic efficiency of the intercropping practices (IPs) compared with monocropping practices (MPs) of sole crops. The yield advantage of IPs in comparison with MPs was assessed using rice equivalent yield (REY) or maize equivalent yield (MEY), land equivalent ratio (LER), area time equivalent ratio (ATER), land use efficiency (LUE), land-equivalent coefficient (LEC), and percentage yield difference (PYD).

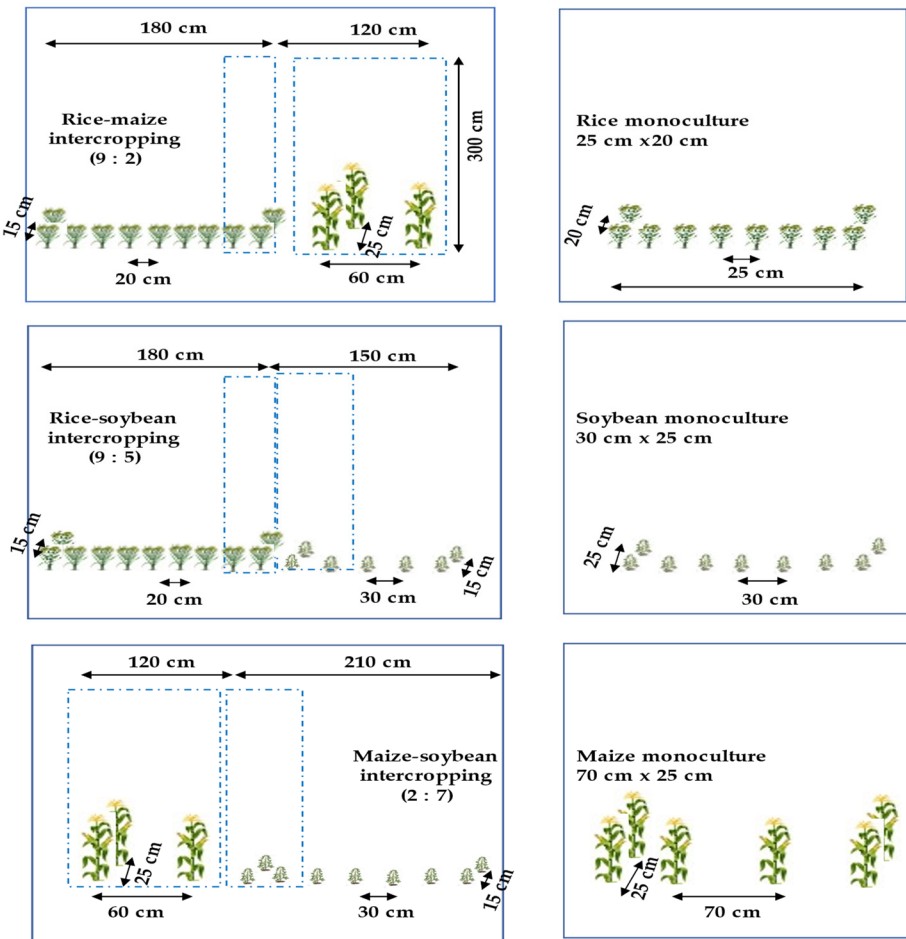

**Figure 3.** Schematic representation of the rice–maize, rice–soybean, and maize–soybean intercropping patterns, and sampling unit (blue dashed lines).

### 2.4.1. Crop Yield Assessment

Rice, maize, and soybean grain yield were measured at physiological maturity from three sampling areas of two rows of rice, maize, or soybean, each 3.0 m long, for each intercropping and sole crop (Figure 3); we chose the net harvest area at random and hand-harvested samples. Grain yields were determined individually for sole crops and intercrops by converting each sampling unit to one hectare. The moisture content of the samples was measured using a seed moisture meter. Grain yield (t ha$^{-1}$) was calculated assuming moisture contents of 14% for rice, 15.5% for maize, and 13% for soybean.

Crop yields of rice, maize, and soybean at different IPs in comparison with MPs were determined based on crop equivalent yield. The crop equivalent yields were REY and MEY. In determining REY and MEY, crop yields are converted into one crop form to allow comparison of the crops cultivated under intercropping. The conversion of the yield is carried out in the form of main crop (rice or maize) equivalent yield by considering the intercrop yield and market price of the main crop and associated intercrops. The market prices of rice, maize, or soybean were based on the farm gate prices in 2019 in each province [33,34]. The REY or MEY expressed in t ha$^{-1}$ were calculated based on the following formulas [29,35].

$$REY \ for \ intercrop = rice \ yield + \frac{associated \ intercrop \ yield \times price \ of \ associated \ intercrop}{rice \ price}$$

$$MEY \ for \ intercrop = maize \ yield + \frac{associated \ intercrop \ yield \times price \ of \ associated \ intercrop}{maize \ price}$$

The yield advantages of intercropping components over standalone cropping systems were evaluated using the total LER. Based on grain yield and the areas each intercrop occupied, the land equivalent ratio was calculated using the following equation [36]:

$$LER = L_a + L_b = \frac{Y_{ia}}{Y_{sa}} + \frac{Y_{ib}}{Y_{sb}}$$

where $L_a$ = partial LER of the main crop (rice or maize) and $L_b$ = partial LER of the associated crop in the intercrop (maize or soybean); $Y_{ia}$ and $Y_{ib}$ are the individual crop yields in intercropping (t ha$^{-1}$), and $Y_{sa}$ and $Y_{sb}$ are their yields as sole crops (t ha$^{-1}$), respectively. The LER is an index that describes the relative land area required to grow the same quantity of both crop species in the mixture if grown as monocultures rather than as mixtures. If the LER is >1, there is a yield advantage.

The ATER provides the yield of intercropping over monocropping in terms of the time taken by component crops in the intercrop [37]:

$$ATER = \frac{\left( \frac{Y_{ia}}{Y_{sa}} \times t_a \right) + \left( \frac{Y_{ib}}{Y_{sb}} \times t_b \right)}{T}$$

where $t_a$ = growth duration (in days) for the main crop, $t_b$ = growth duration (in days) for the associated crop, $T$ = growth duration (in days) for the intercropping system.

Since the LER value is often overestimated and the ATER value underestimated, they were averaged to provide a more accurate estimate of LUE. By utilizing values of LER plus ATER, the LUE percentage was determined according to [38] as follows:

$$LUE = \frac{LER + ATER}{2} \times 100$$

The LEC is a measure of the proportion of the yield in one intercropping component explained by the presence of the other component and was calculated [39] as:

$$LEC = L_a \times L_b$$

The PYD is defined as the yield difference between the sole crop at full population and the intercrop expressed as a percentage. The PYD between the monoculture and the intercropping was calculated [40] as:

$$PYD = 100 - \left[ \frac{Y_{sa} - Y_{ia}}{Y_{sa}} + \frac{Y_{sb} - Y_{ib}}{Y_{sb}} \right] \times 100$$

where $Y_{sa}$ and $Y_{sb}$ are the yields as sole crops (t ha$^{-1}$); $Y_{ia}$ and $Y_{ib}$ are the individual crop yields in intercropping (t ha$^{-1}$), respectively. We considered the monoculture yield as 100%.

### 2.4.2. Crop Competition Assessment

The aggressiveness index (AI) was adopted as a competitive index to measure the extent to which the relative yield of one crop in the mixture was higher than that of the other. AI was used to evaluate the competitive relationship between two crops in the intercropping systems, as suggested by [41], using the following formula:

$$AI_a = \left( \frac{Y_{ia}}{Y_{sa} \times Z_{ia}} \right) - \left( \frac{Y_{ib}}{Y_{sb} \times Z_{ib}} \right)$$

$$AI_b = \left( \frac{Y_{ib}}{Y_{sb} \times Z_{ib}} \right) - \left( \frac{Y_{ia}}{Y_{sa} \times Z_{ia}} \right)$$

where $AI_a$ = the aggressivity of the main crop (rice or maize); $AI_b$ = the aggressivity of the associated intercrop (maize or soybean); $Z_{ia}$ is the sown proportion of the main crop in

intercropping with the associated crop; and $Z_{ib}$ is the sown proportion of the associated crop in intercropping. If $AI_a$ or $AI_b = 0$, then both crops in the intercropping system are equally competitive. If $AI_a$ is positive, this denotes the dominance of the main crop over the associated crop in the intercrop, whereas when it is negative, this indicates that the associated crop is the dominant species.

The competitive ratio (CR) was used to assess the competitive ability of the component crops in an intercropping system. The CR represents simply the ratio of the individual LERs of the two component crops [35,42]:

$$CR_a = \frac{LER_a}{LER_b} \times \frac{Z_{ia}}{Z_{ib}}$$

$$CR_b = \frac{LER_b}{LER_a} \times \frac{Z_{ib}}{Z_{ia}}$$

where $CR_a$ = the competitive ratio of the main crop (rice or maize); $CR_b$ = the competitive ratio of the associated crop in the intercrop (maize or soybean). When $CR_a < 1$, there is a positive benefit of intercropping, suggesting that the main crop can be grown in association with the associated crop, whereas when $CR_b > 1$, this is an indication of a negative benefit. If the difference between $CR_a$ and $CR_b$ is 0, then the main crop and the companion crop are equally competitive. However, if subtracting $CR_b$ from $CR_a$ gives a positive value, then the main crop in the intercrop is dominant. In contrast, a negative value indicates that the companion crop dominates the main crop [43].

2.4.3. Economic Efficiency Assessment

We performed an economic analysis to determine each intercropping system's financial viability. The production facilities are all items purchased for field production, such as seeds, fertilizers, and pesticides. Paid workers include all contractual costs such as land preparation, planting, and harvesting. The variable cost denotes the total expenses of production field purchases plus paid workers according to the farm operational activities for each system and required inputs. The cost of production is the total that a farmer's household spends on cultivating one hectare of MPs and IPs per planting season. The costs only cover activities involved in production of standard products. They include imputation of cost items, not in absolute terms, spent by households as a rent estimation for property (land, equipment, and unpaid family workers) used in crop cultivation that is owned by households. We calculated the total expenses for the production of MPs and IPs based on local rates. Revenues were calculated by multiplying the grain yield of upland rice, maize, or soybean crops by the farm gate price in 2019 for each province. The gross margin was determined by subtracting the revenues from the total variable cost. The revenue cost ratio was calculated by dividing revenues by total production cost. Profit was determined by subtracting the revenues from the total production cost. The economic data were converted into USD using an exchange rate of USD 1 = IDR 14,250.

The yields of all crops in different intercropping systems and in the sole cropping systems along with their economic returns were analyzed using intercropping advantage (IA), the income equivalent ratio (IER), the monetary advantage index (MAI), and gross margin (GM) analysis.

Intercropping advantage (IA) was used as a gauge for the economic viability of the intercropping system. This index is derived from [44,45]:

$$IA = IA_a + IA_b$$
$$IA_a = AYL_a \times P_a$$
$$IA_b = AYL_b \times P_b$$

where $IA_a$ is the intercropping advantage of the main crop (rice or maize); $IA_b$ is the intercropping advantage of the associated crop in the intercrop (maize or soybean). $P_a$ is

the commercial value of the main crop; $P_b$ is the commercial value of the associated crop in the intercrop.

The monetary advantage index (MAI) is an important index for determining the economic viability of intercropping. MAI describes the financial advantage of intercropping compared to monoculture. A higher MAI value indicates that this cropping system is more profitable than another. It was calculated according to [46,47] as:

$$MAI = \frac{(value\ of\ combined\ intercrops) \times (LER - 1)}{LER}$$

The income equivalent ratio (IER) is the relative land area required under a sole crop to acquire the equivalent gross income as received from 1 ha of intercropping at the same management level. It comprises the conversion of LER in terms of economic returns. IER > 1 denotes the intercropping system is advantageous. The formula for IER given by [48,49] is as follows:

$$IER = \frac{GI_{ia}}{GIsa} + \frac{GIib}{GIsb}$$

where $GI_{ia}$ and $GI_{ib}$ are the individual gross incomes for intercropping (USD ha$^{-1}$), and $GI_{sa}$ and $GI_{sb}$ are their gross incomes (USD ha$^{-1}$) as sole crops, respectively.

In recent years, studies have increasingly compared net returns, as measured by the activity gross margin (GM), between intercropping and monocropping systems [24,50]. An economic advantage of intercropping exists if the GM from intercropping is higher than that of monocropping [51,52]. To determine the system economic gain per hectare, we used a GM analysis model that is equal to the difference between total revenue (TR) and total variable cost (TVC) and is expressed as follows:

$$GM\ (\pi) = \sum TR - \sum TVC$$

The TR means the total market price of production per hectare multiplied by the crop yields, and TVC includes input costs such as seeds, fertilizers, pesticides, and paid labor.

### 2.5. Data Analysis

We compared treatment means in the statistical analysis [53]. The least significant difference (LSD) at a five percent level was applied to examine the significance of differences in equivalent grain yield, gross margin, profit, production cost, and cost of paid workers between the IPs of rice–maize, rice–soybean, or maize–soybean and MPs of sole crop treatment means.

## 3. Results

### 3.1. Grain Yields of IP and MP

Grain yields under IPs (rice–maize, rice–soybean, and maize–soybean) showed significantly higher yields per cycle than under MPs (sole rice or sole maize) in the rainfed areas of western Indonesia with a wet climate (Table 3). On average, the REY of rice–maize intercropping indicated an increase in rice productivity of 2.92 ± 0.04 t ha$^{-1}$ or 78.2 ± 0.87% higher than the sole rice crop. Under rice–soybean intercropping, the average mean grain yield of REY showed an increase in rice productivity of 2.35 ± 0.01 t ha$^{-1}$ or 58.4 ± 0.35% higher than the sole rice crop. Compared to sole maize, the MEY of maize–soybean intercropping increased maize productivity per cycle by 2.89 ± 0.05 t ha$^{-1}$, or 66.8 ± 0.81%.

**Table 3.** Yield increase of rice equivalent yield (REY) and maize equivalent yield (MEY) for different intercropping practices (IPs) in comparison with monocropping practices (MPs) on rainfed areas of western Indonesia with a wet climate, 2019.

| Intercropping | Mean IPs | | | Mean MPs | | | REY | MEY | Yield Increase | |
|---|---|---|---|---|---|---|---|---|---|---|
| | Rice | Maize | Soybean | Rice | Maize | Soybean | | | | |
| | (t ha$^{-1}$) | | | | | | (t ha$^{-1}$) | | (t ha$^{-1}$) | (%) |
| Rice–maize | 2.926 | 2.839 | – | 3.724 b * | 4.368 | – | 6.646 a | – | 2.922 | 78.2 |
| | ±0.030 | ±0.045 | – | ±0.021 | ±0.020 | – | ±0.055 | – | ±0.040 | ±0.865 |
| | 0.449 ** | 0.668 | – | 0.318 | 0.299 | – | 0.828 | – | 0.595 | 12.764 |
| Rice–soybean | 3.155 | – | 1.648 | 4.035 b * | – | 2.181 | 6.381 a | – | 2.346 | 58.4 |
| | ±0.011 | – | ±0.011 | ± 0.017 | – | ±0.059 | ±0.020 | – | ±0.010 | ±0.354 |
| | 0.159 ** | – | 0.167 | 0.259 | – | 0.230 | 0.298 | – | 0.157 | 5.317 |
| Maize–soybean | – | 3.407 | 1.709 | – | 4.353 b * | 2.025 | – | 7.327 a | 2.885 | 66.8 |
| | – | ±0.026 | ±0.008 | – | ±0.024 | ±0.005 | – | ± 0.036 | ±0.028 | ±0.807 |
| | – | 0.395 ** | 0.113 | – | 0.362 | 0.071 | – | 0.536 | 0.426 | 12.102 |

\* Means followed by the same letter within rows are not significantly different at *p* < 0.05; \*\* Std Dev.; ± denotes the standard error of the 15 replications; – not applicable.

### 3.2. Yield Advantage

Analyses of the LER, ATER, LUE, LEC, and PYD values for different IPs in comparison with MPs are presented in Table 4. The results show that the mean LER and ATER values for rice–maize, rice–soybean, and maize–soybean intercropping are above one in all intercropping systems. Using the LER to describe the magnitude of the yield increase with intercropping over sole cropping, we found that the yield advantages of the IPs were 44% for rice–maize, 54% for rice–soybean, and 63% for maize–soybean compared to MPs. On average, ATER values were 32%, 30%, and 30% higher in rice–maize, rice–soybean, and maize–soybean intercropping systems, respectively, compared to sole cropping. The LUE of rice–maize, rice–soybean, and maize–soybean intercropping systems also increased by 38%, 43%, and 46%, respectively, compared to sole cropping. Intercropping increased the equivalent land coefficient (LEC) of rice–maize, rice–soybean, and maize–soybean by 0.50, 0.60, and 0.63. The PYD showed a similar trend with LER in all the intercropping systems. All indices indicate that rice–maize, rice–soybean, and maize–soybean intercropping are more beneficial than sole cropping of rice, maize, and soybean.

**Table 4.** Land equivalent ratio (LER), area time equivalent ratio (ATER), land use efficiency (LUE), land equivalent coefficient (LEC), and percentage yield difference (PYD) for different IPs in comparison with MPs on rainfed areas of western Indonesia with a wet climate.

| Intercropping | LER | ATER | LUE (%) | LEC | PYD (%) |
|---|---|---|---|---|---|
| Rice–maize | 1.44 ± 0.009 0.136 * | 1.32 ± 0.034 0.131 | 138.22 ± 0.883 13.245 | 0.50 ± 0.007 0.105 | 43.83 ± 0.907 13.607 |
| Rice–soybean | 1.54 ± 0.006 0.096 * | 1.30 ± 0.026 0.101 | 142.70 ± 0.482 7.224 | 0.60 ± 0.019 0.074 | 53.88 ± 0.612 9.183 |
| Maize–soybean | 1.63 ± 0.007 0.107 * | 1.30 ± 0.018 0.070 | 145.78 ± 0.451 6.772 | 0.63 ± 0.022 0.086 | 62.94 ± 0.711 10.616 |

\* Std Dev; ± denotes the standard error of the 15 replications.

### 3.3. Crop Competition

The crop competition in intercropping was measured using competition functions such as the aggressiveness index (AI) and competitive ratio (CR) (Table 5). In rice–maize intercropping, both rice and maize had positive aggressivity, indicating that both crops were the dominant species in the intercropping system. Under rice–soybean intercropping, rice had positive aggressivity, indicating that rice was the dominant crop. In maize–soybean

intercropping, maize had positive aggressivity, showing that maize was aggressive or dominant in the intercropping system.

**Table 5.** Estimated values of aggressiveness index (AI) and competitive ratio (CR) for different IPs in comparison with MPs on rainfed areas of western Indonesia with a wet climate.

| Intercropping | Aggressivity | | | Competitive Ratio | | |
|---|---|---|---|---|---|---|
| | $AI_{rice}$ | $AI_{maize}$ | $AI_{soybean}$ | $CR_{rice}$ | $CR_{maize}$ | $CR_{soybean}$ |
| Rice–maize | 1.54 ± 0.014 | 0.30 ± 0.033 | – | 1.96 ± 0.040 | 0.56 ± 0.012 | – |
| | 0.206 * | 0.496 | – | 0.604 | 0.186 | – |
| Rice–soybean | 0.18 ± 0.014 | – | −0.26 ± 0.010 | 0.98 ± 0.007 | – | 1.03 ± 0.007 |
| | 0.205 * | – | 0.157 | 0.106 | – | 0.110 |
| Maize–soybean | – | 1.62 ± 0.013 | −0.86 ± 0.017 | – | 0.61 ± 0.010 | 1.76 ± 0.038 |
| | – | 0.201 * | 0.257 | – | 0.147 | 0.571 |

± denotes the standard error of the 15 replications; * Std Dev.; – not applicable.

The results for the CR values were in accordance with those for the A values. The value of the $CR_{rice}$ under rice–maize intercropping where rice was the main crop is 1.96, and the value for $CR_{maize}$ as the associated crop is 0.56. Since $CR_{rice}$ is >1, there is a negative benefit of intercropping, suggesting that rice as the main crop should not be grown in association with maize as an associated crop. However, the rice–soybean and maize–soybean intercropping values of $CR_{rice}$ and $CR_{maize}$ as the main crop are <1. This means there is a positive benefit of intercropping, suggesting that rice or maize as the main crop may be grown in association with the soybean crop.

*3.4. Economic Efficiency*

The economic efficiency, calculated as the income equivalent ratio (IER), intercropping advantage (AI), and monetary advantage index (MAI) for each cropping system are shown in Table 6. The partial $IER_{rice}$ value was higher in rice than in maize under rice–maize intercropping. However, under rice–soybean and maize–soybean intercropping, partial $IER_{soybean}$ was higher than $IER_{rice}$ or $IER_{maize}$. The total intercropping advantage follows the same trend as the total IER. The total IER and AI were higher for maize–soybean, followed by rice–soybean and then rice–maize intercropping. In all cropping pattern systems, total IER and AI values were >1.

**Table 6.** The partial income equivalent ratio (IER) for rice, maize, and soybean, and total IER for each cropping system. Intercropping advantage (AI) for rice, maize, and soybean; total AI for each cropping system; and monetary advantage index (MAI) at different IPs in comparison with MPs on the rainfed areas of western Indonesia with a wet climate.

| Intercropping | Income Equivalent Ratio | | | | Intercropping Advantage | | | | MAI |
|---|---|---|---|---|---|---|---|---|---|
| | $IER_{rice}$ | $IER_{mazie}$ | $IER_{soybean}$ | Total | $IA_{rice}$ | $IA_{mazie}$ | $IA_{soybean}$ | Total | USD $h^{-1}$ |
| Rice–maize | 1.46 ± 0.01 | 1.36 ± 0.02 | – | 2.82 ± 0.02 | 99.3 ± 5.17 | 228.6 ± 6.02 | – | 327.9 ± 6.3 | 581 ± 13.4 |
| | 0.18 * | 0.23 | – | 0.36 | 77.55 | 90.27 | – | 94.99 | 100.78 |
| Rice–soybean | 1.54 ± 0.03 | – | 1.78 ± 0.01 | 3.32 ± 0.03 | 197.6 ± 1.59 | – | 202.9 ± 4.10 | 400.4 ± 4.1 | 633 ± 6.2 |
| | 0.43 * | – | 0.17 | 0.46 | 23.89 | – | 61.57 | 61.41 | 92.99 |
| Maize–soybean | – | 1.63 ± 0.02 | 1.79 ± 0.03 | 3.42 ± 0.04 | – | 435.4 ± 3.92 | 26.4 ± 6.21 | 461.7 ± 4.7 | 692 ± 6.9 |
| | – | 0.24 * | 0.45 | 0.55 | – | 55.77 | 93.22 | 70.65 | 103.10 |

± denotes standard error (n = 15) of the mean; * Std. dev.; – not applicable.

Intercropping of rice–maize, rice–soybean, and maize–soybean also consistently affected the MAI. MAI values were positive in all the intercropping systems and were higher than one. MAI was higher in maize–soybean (USD 692 ha$^{-1}$), followed by rice–soybean (USD 633 ha$^{-1}$) and then rice–maize intercropping (USD 581 ha$^{-1}$).

### 3.5. Gross Margin and Profit Analysis

The economic performances of rice–maize, rice–soybean, and maize–soybean IPs in comparison with MPs were calculated based on total production cost, total revenues, revenue cost ratio, gross margin, and profit (Table 7). Demonstration plots used in the farmers' fields in the rainfed areas with a wet climate of western Indonesia indicated that farmers spent 82.9% and 69.0%, respectively, more on paid workers when comparing rice–maize and rice–soybean intercropping with sole rice. Farmers spent 16.6% more on paid workers when comparing maize–soybean intercropping with sole maize. Farmers used HYVs; thus, farmers spent more on buying seeds, fertilizers, and pesticides. Higher costs of paid workers and production field purchases resulted in higher total variable costs in IPs compared to MPs. However, since REY and MEY under IPs indicated significantly higher yields per cycle than under the MPs, IPs resulted in a substantially lower cost of production $t^{-1}$ and a lower cost of paid workers $t^{-1}$. Cost of production $t^{-1}$ of rice was USD 211 under rice–maize intercropping compared with USD 229 $t^{-1}$ for sole rice. The cost of rice production was USD 188 $t^{-1}$ under rice–soybean intercropping compared with USD 228 $t^{-1}$ under sole rice. The cost of production of maize under maize–soybean intercropping was USD 190 $t^{-1}$ compared with USD 225 for sole maize. The GMs of IPs of rice–maize, rice–soybean, and maize–soybean were significantly higher than the MPs of rice, maize, or soybean as sole crops. Thus, IPs provided a significantly higher profit and net return than growing one crop alone. There were additional net return gains of USD 160 ha$^{-1}$ (47.1%) and USD 203 ha$^{-1}$ (57.3%) per cycle under rice–maize and rice–soybean intercropping compared with MP. Maize–soybean intercropping produced an additional net return gain of USD 153 (62.5%) compared with MP.

**Table 7.** Profit analysis of the rice–maize, rice–soybean, and maize–soybean IPs in comparison with MPs in rainfed areas of western Indonesia with a wet climate.

| Item | IPs | St D | Rice | St D | Maize | St D | Soybean | St D |
|---|---|---|---|---|---|---|---|---|
| | (USD ha$^{-1}$) | | | | (USD h$^{-1}$) | | | |
| | | | | Rice–maize | | | | |
| Production field purchases | 234 ± 0.8 | 12.6 | 78 ± 0.2 | 9.2 | 214 ± 0.5 | 16.8 | – | – |
| Paid workers | 264 ± 1.2 | 17.5 | 145 ± 0.7 | 10.2 | 224 ± 0.7 | 10.8 | – | – |
| Total variable cost | 498 ± 1.5 | 22.2 | 224 ± 0.7 | 16.5 | 550 ± 1.5 | 21.8 | – | – |
| Total production cost | 1376 ± 6.6 | 98.5 | 846 ± 3.1 | 46.1 | 988 ± 1.5 | 51.9 | – | – |
| Total revenue | 1902 ± 88.1 | 271.2 | 1212 ± 61.2 | 167.8 | 1466 ± 50.1 | 75.7 | – | – |
| Revenue cost ratio | 1.39 ± 0.1 | 0.25 | 1.43 ± 0.1 | 0.2 | 1.5 ± 0.0 | 0.1 | – | – |
| Gross margin | 1404 ± 18.7 a * | 280.5 | 989 ± 10.7 c | 160.6 | 1027 ± 4.9 b | 73.7 | – | – |
| Profit | 525 ± 20.3 a | 74.6 | 366 ± 9.8 c | 47.2 | 477 ± 4.6 b | 69.0 | – | – |
| Cost of production $t^{-1}$ ** | 211 ± 2.2 b | 34.9 | 229 ± 1.3 a | 89.6 | 227 ± 1.1 a | 36.7 | – | – |
| Cost of paid workers $t^{-1}$ | 40 ± 0.4 b | 6.2 | 39 ± 0.2 b | 5.0 | 52 ± 0.3 a | 8.3 | – | – |
| Profit increase to sole rice | 160 ± 16.3 (47.1% ± 5.0) | 22.8 | – | – | – | – | – | – |
| | | | | Rice–soybean | | | | |
| Production field purchases | 194 ± 0.4 | 16.3 | 81 ± 0.2 | 12.2 | – | – | 116 ± 0.3 | 14.8 |
| Paid workers | 264 ± 2.1 | 31.1 | 157 ± 0.8 | 11.4 | – | – | 104 ± 0.22 | 13.3 |
| Total variable cost | 458 ± 2.4 | 35.3 | 237 ± 0.7 | 20.7 | – | – | 220 ± 0.37 | 15.6 |
| Total production cost | 1194 ± 3.3 | 50.0 | 917 ± 1.2 | 47.5 | – | – | 634 ± 0.46 | 56.8 |
| Total revenue | 1811 ± 9.6 | 144.5 | 1331 ± 10.4 | 156.1 | – | – | 1023 ± 7.87 | 118.0 |
| Revenue cost ratio | 1.52 ± 0.1 | 0.14 | 1.45 ± 0.1 | 0.17 | – | – | 1.62 ± 0.01 | 0.19 |
| Gross margin | 1353 ± 9.8 a * | 146.5 | 1093 ± 10.4 b | 155.6 | – | – | 803 ± 7.85 c | 117.8 |
| Profit | 617 ± 10.2 a | 52.7 | 414 ± 10.3 b | 55.1 | – | – | 390 ± 7.98 c | 39.7 |
| Cost of production $t^{-1}$ ** | 188 ± 1.0 c | 26.1 | 228 ± 1.0 b | 15.0 | – | – | 293 ± 1.94 a | 29.1 |
| Cost of paid workers $t^{-1}$ | 42 ± 0.4 b | 5.9 | 39 ± 0.3 b | 4.4 | – | – | 48 ± 0.31 a | 4.7 |
| Profit increase to sole rice | 203 ± 6.7 (57.3% ± 2.2) | 24.1 | – | – | – | – | – | – |

**Table 7.** *Cont.*

| Item | IPs | St D | Rice | St D | Maize | St D | Soybean | St D |
|---|---|---|---|---|---|---|---|---|
| | **(USD ha$^{-1}$)** | | | | **(USD h$^{-1}$)** | | | |
| | | | | Maize–soybean | | | | |
| Production field purchases | 222 ± 2.9 | 43.8 | – | – | 207 ± 0.3 | 5.2 | 116 ± 0.32 | 4.8 |
| Paid workers | 256 ± 1.5 | 22.3 | – | – | 221 ± 0.9 | 13.0 | 104 ± 0.22 | 3.4 |
| Total variable cost | 478 ± 2.9 | 44.2 | – | – | 428 ± 1.1 | 16.0 | 220 ± 0.37 | 5.6 |
| Total production cost | 1371 ± 9.9 | 147.8 | – | – | 975 ± 5.3 | 79.3 | 634 ± 0.46 | 6.8 |
| Total revenue | 1795 ± 7.1 | 106.8 | – | – | 1246 ± 7.2 | 108.6 | 976 ± 7.39 | 110.8 |
| Revenue cost ratio | 1.32 ± 0.1 | 0.13 | – | – | 1.28 ± 0.0 | 0.12 | 1.54 ± 0.01 | 0.17 |
| Gross margin | 1317 ± 7.4 a * | 111.7 | – | – | 819 ± 7.1 b | 106.4 | 756 ± 7.32 c | 109.9 |
| Profit | 425 ± 10.2 a | 52.9 | – | – | 272 ± 7.2 c | 37.2 | 343 ± 7.26 b | 48.9 |
| Cost of production t$^{-1}$ ** | 190 ± 1.6 c | 21.7 | – | – | 225 ± 1.7 b | 25.2 | 313 ± 0.74 a | 11.1 |
| Cost of paid workers t$^{-1}$ | 36 ± 0.3 b | 6.8 | – | – | 51 ± 0.3 a | 4.8 | 51 ± 0.13 a | 1.9 |
| Profit increase to sole maize | 153 ± 6.3 (62.5% ± 2.4) | 20.5 | – | – | – | – | – | – |

\* Means followed by the same letter within rows are not significantly different at $p < 0.05$; ± denotes standard error (n = 15) of the mean; – not applicable; \*\* cost of production t$^{-1}$ for each intercropping was calculated by dividing the total production cost (USD ha$^{-1}$) by REY (t ha$^{-1}$) for rice–maize intercropping and rice–soybean intercropping, and MEY (t ha$^{-1}$) for maize–soybean intercropping as shown in Table 3.

## 4. Discussion

### 4.1. Yield Advantage of the Intercropping Systems

In our demonstration plots, the mean REY significantly increased for rice productivity under rice–maize and rice–soybean intercropping compared with the sole rice crop. However, the productivity of rice under MP of rice–maize and rice–soybean intercropping varies from 3.7 to 4.0 t ha$^{-1}$ (Table 3). These yields were not optimum due to small-scale farmers on rainfed land with a wet climate using lower N doses (45 kg N ha$^{-1}$) than the blanket recommendation dose (67.5 kg N ha$^{-1}$). The high-yielding variety using Inpago-12 is Al-toxicity tolerant and only resistant to blast race 033 of *Pyricularia grisea* [54], which is dominant in Java but not outside of Java. Higher N application in high humidity conditions would increase the infestation of blast disease, especially neck blast [55]. The HYV Inpago-12 has an average grain yield obtained from adaptability test activities, representing the agroecological characteristics of rainfed rice production center areas of 6.7 t ha$^{-1}$ [54].

The MEY under maize–soybean intercropping was considerably higher than that for the sole maize crop. Intercropping gave greater combined yields than those obtained from either crop grown alone due to more efficient and complementary use of available growth resources [56,57]. The findings of other researchers regarding the outcome of cereal–cereal intercrops have shown that when water is not limited, wheat–maize intercropping can increase grain yields by 26 to 64% compared with the corresponding sole cropping [58]. However, there is interspecific competition in wheat–maize intercropping under water stress [59].

The mean LER and ATER values were above one in all the intercropping systems. We found that the LER value was 1.44 for rice–maize, 1.54 for rice–soybean, and 1.63 for maize–soybean. Some researchers have shown LER values of 1.94 under maize–soybean intercropping in China [43], 1.55 under rice–maize intercropping in Uganda [60], and 1.60 for rice–soybean intercropping in Japan [61].

However, in rice–maize intercropping, both rice and maize had positive aggressivity, indicating that both crops were the dominant species in the intercropping system. Thus, the CR value followed the A value. The value of the CR ratio under rice–maize intercropping wherein rice is the main crop was 1.96, and the value of CR maize as the associated crop was 0.56. Since CR rice was >1, there is a negative benefit of intercropping, suggesting that rice as the main crop should not be grown in association with maize as an associated crop [43,62].

Based on LER, the yield advantages of the IPs compared with MPs were the highest for maize–soybean (63%), followed by rice–soybean (54%) and then rice–maize (44%). These, in line with the MAI values, were higher in maize–soybean (USD 692 ha$^{-1}$), followed by rice–soybean (USD 633 ha$^{-1}$) and rice–maize intercropping (USD 581 ha$^{-1}$). These values imply that maize–soybean is more economically viable to intercrop than rice–soybean and rice–maize intercropping under rainfed conditions with a wet climate in western Indonesia.

### 4.2. Economic Performance of the Intercropping Systems

The economic analysis of the demonstration plots used in the rainfed conditions of farmers' fields without mechanization provides the first original findings regarding cereal and cereal–soybean intercropping systems using calculated MAI, IA, IER, GM, and net returns values in Southeast Asia. Crop yields, profit gains, and labor requirements are essential to accelerate the promotion of intercropping systems for smallholder farmers under farmer participatory demonstration plots [21,63,64]. Total variable costs were higher for intercropping systems than for monocropping practices. Farmers used high-yielding varieties in all the intercropping patterns, so the production costs of rice–maize, rice–soybean, and maize–soybean intercropping remained high due to the increased costs related to expenses for the purchase of seeds, fertilizers, pesticides, and paid workers, especially for planting and yield processing of the intercrop. The fixed cost was also higher for IPs than MPs because households spend more for IPs than MPs on equipment and maximizing unpaid family workers by adding paid workers used in crop cultivation. The increased workload and therefore expenses may also be a reason why farmers do not grow two crops at the same time, although such a practice could minimize unemployment labor, considering that farming is not practiced throughout the year.

One disadvantage of intercropping is the higher maintenance cost, particularly planting, weeding, and harvesting, which may have to be done manually. This is not a severe problem in areas where excess farm labor is cheap, for example, in Sumatra and Kalimantan. For areas close to the industrial areas with better soil fertility, such as in Java and Bali, where farm labor is scarce and expensive, intercropping will result in increased costs. Thus, small-scale farmers lacking mechanization will get more benefits from monocropping systems.

However, intercropping resulted in a significantly higher GM compared to monocropping. An economic advantage of intercropping exists if the GM from intercropping is considerably higher than that of monocropping, thus increasing the financial benefit for farmers [51,64]. Intercropping of rice–maize, rice–soybean, or maize–soybean significantly increased total crop productivity and resulted in additional net return gains of USD 160 (47.13%), USD 203 (57.28%), and USD 153 ha$^{-1}$ (62.47%) per cycle, respectively, for IPs compared with MPs.

Increased yield productivity under IPs reduces the cost of production and increases the productivity of paid workers for the main crop compared with just rice or maize. In terms of achieving food self-sufficiency, an archipelagic country such as Indonesia has a natural weakness. As an island country, there are relatively high costs for transporting production facilities and output that are needed to cover regional production gaps. For example, 55.5% of rice, 54.1% of maize, and 62.3% of soybean were produced on Java Island [14]. Therefore, land productivity must increase so that household food security in each region becomes more evenly distributed, which will reduce transportation costs between islands/region; and the proportion of farmers' household expenditure for own consumption reaches an average of 60 % of total spending [65]. Other islands outside Java, such as Kalimantan, Sulawesi, and Papua, can reduce dependence on food supply from production centers in Java; both meet the nutritional needs of families and animal feed. We hope the agronomic and economic advantages of intercropping will stimulate trading in each region.

Recent research indicates there is more optimal use of the available environmental factors in intercropping of rice or maize than in a monoculture [46,59,66]. Rice–soybean or maize–soybean combinations provide food and nutritional security to smallholders of

rainfed areas. They may therefore be considered suitable options in ensuring small farmers' food and livelihood security.

## 5. Conclusions

Based on different indices used to assess crop yields, competition intensity, and economic efficiency, our results demonstrate that intercropping practices are more beneficial than monocropping practices in the compared systems. The results for intercropping indicate that maize–soybean provides the highest percentage of net return gains (62.5%) followed by rice–soybean (57.3%), with the lowest for rice–maize (47.1%). The intercropping advantage assessment supports the estimation of economic indices. In using the LER to describe the magnitude of the yield increase for IP over MP, yield advantages of 63%, 54%, and 44% were recorded for maize–soybean, rice–soybean, and rice–maize, respectively. ATER values were 30%, 30%, and 32% higher for the same IP compared with sole cropping. The LUE values also increased by 46%, 43%, and 38%, respectively, in comparison to sole cropping. Intercropping increased the LEC of maize–soybean, rice–soybean, and rice–maize by 0.66, 0.60, and 0.50, respectively. The PYD showed a similar trend to LER in all intercropping systems. The goals of smallholders in rainfed areas are generally to reduce risks, generate income, and achieve food security. These goals, to some extent, could be achieved by practicing intercropping. Given the above data and conclusions, farmers could more effectively utilize their land area and time and thus increase their income and production efficiency by practicing intercropping, especially if adopting maize–soybean and rice–soybean intercropping systems.

**Author Contributions:** All authors contributed equally. Conceptualization, E.E.; Data curation, S.S. (Susilawati Susilawati), S.S. (Slameto Slameto), N.M.D.R., F.D.A., J.J., A.F. and A.B.; Formal analysis, N.M.D.R.; Investigation, S.S. (Susilawati Susilawati), S.S. (Slameto Slameto), N.M.D.R., F.D.A., J.J. and A.F.; Methodology, S.S. (Slameto Slameto) and A.J.; Software, A.B.; Validation, E.E., A.B., A.J. and H.S.; Writing—original draft, E.E.; Writing—review and editing, H.S. All authors have read and agreed to the published version of the manuscript.

**Funding:** This research was funded by national core budget (018) G 521211. We are grateful for the financial support from the Indonesian Agency for Agricultural Research and Development (IAARD), Ministry of Agriculture of the Republic of Indonesia.

**Data Availability Statement:** Not applicable.

**Acknowledgments:** We would like to thank the reviewers for their comprehensive and beneficial comments and suggestions that helped us to improve the manuscript.

**Conflicts of Interest:** The authors declare no conflict of interest.

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
