# Peer review of "Yield Advantage and Economic Performance of Rice–Maize, Rice–Soybean, and Maize–Soybean Intercropping in Rainfed Areas of Western Indonesia with a Wet Climate"

_agronomy, doi:10.3390/agronomy12102326_

Round 1

Reviewer 1 Report

Make clear from the start that you are comparing intercropping and monocropping by small scale farmers without mechanisation. In the history of agriculture, intercropping has been a normal practice where production for self-sufficiency dominated. Monocropping appeared under the influence of market oriented mechanized farming. 

Consider limiting yourself as far as the number of criteria used for demonstrating the agronomic and economic (dis)advantages of intercropping versus monocropping, to be able to capture and maintain the interest of the readers. If you consider all of importance, you should treat them individually and proof the interest of measuring or calculating them all.

A key result is missing! You mention in the introduction that monocropping dominates strongly (r. 58 - 60), but insist through your research that it has serious agronomic and economic disadvantages. Nowhere you help understanding this contradiction. Nowhere you help understanding why small scale farmers lacking mechanization are going for monocropping.

par. 1

One expects in this introduction information about the known reasons why intercropping could have advantages. In the context of this study at least i.) the potential advantage of combining a leguminous crop with cereals, and ii.) the (shorter) length of the growing cycle versus the growing season could have been mentioned. Your hypothesis (r. 91 - 93) will become better underpinned.

A contradiction exists between r. 43 & 44 and r. 82 - 84

r. 49 - 51 This suggests that farmers yields are close to the potential rainfed yields; high fertilizer doses are combined with high yielding varieties and pesticide use. Why much lower doses of fertilizers are used in the experiment (r. 151 - 155).

par. 2.2

r. 138 & 139 is not clear

r. 151 - 157: You are looking for improving food security through change in agricultural practices. In this context, it is not logic using the locally used doses. You hope changing what farmers can afford.

Explain why farmers use on sole rice much less fertilizer than on sole maize. By the way, it is difficult to believe that 45 kg/ha of can lead to the MP rice yields of table 4.

Table 3 and fig. 2: Explain why using different plant densities in MP's and IP's

par. 2.4.1; r. 175 - 178: Why measuring moisture content while using commercial standards?

par. 2: A list of all abbreviations used will be useful!

Table 4: What is the origin of the yield increase of 2.885 t/ha in case of Maize-soybean?

par. 2.4.3: It will be easier to understand the results (par. 3.4 & 3.5) if information about the costs is more elaborated. Information about the fixed costs is almost lacking.

Table 8 (and several other tables):

How is it possible to obtain such small Std. dev. in view of the limited number of fields in very different regions?

Reading of table difficult by presenting even the $cents.

Make clear why the fixed costs are so different between IPs & MPs.

In view of your target group, small scale farmers, one expects information about the fraction of production that has to be used for own consumption. 

Why the information below the table? Already presented in text on several places.

par. 4: Rather meager. Among others in view of all agronomic and economic criteria used for comparing IPs and MPs, one expects more information about practices that could be applied to obtain the highest possible yields, income .... Even the variation between farmers and regions could be used in this context. Not only the average measurements obtained are of interest.

Why such a limited proportion of farmers adopted IPs (r. 58 - 61)?

r. 468 - 476: Is the difference between agronomic and economic advantages of IPs enough to make it interesting trading and transporting food between regions?  

Author Response

We want to thank the reviewer for your comprehensive and beneficial comments and suggestions that helped us to improve the manuscript. Attached, please find an explanation statement from us and the revised manuscript based on observations and recommendations from Reviewer-1.

Reviewer 2 Report

1General comment:

Data presented in this manuscript were collected from farms which are basically heterogeneous, authors decided to pull them together in the analysis and generalized the discussion, conclusion and recommendation, Any reason(s) for that?

Areas to address

  Line 14; delete aimed to; change compared to “compared”

2.       Line 15 delete words…in terms of their…..replace them with “for”

3.       The abstracts and MM sections mention you have worked with 15 cooperating farmers per cropping treatment, where they equally distribute per district and villages? How were they selected?

4.       What does No. column in Table 1 mean?

5.       Weather data presented in Table 2 are averages and do not tell much what happened in the season. These can better be presented in Figure as cumulative daily (or decade) rainfall across the season or a cumulative graph (for Rainfall) and daily for (temperature). Other weather information can be left out.  

6.       Table 3; avoid using many lines in the Table

7.       Line 122; delete the word farm before pool.

Author Response

We want to thank the reviewer for your comprehensive and beneficial comments and suggestions that helped us to improve the manuscript. Attached, please find an explanation statement from us and the revised manuscript based on observations and recommendations from Reviewer-2.
